# On the Use of a Hydrogen-Fueled Engine in a Hybrid Electric Vehicle

## Stefano Beccari

Department of Engineering, University of Palermo, Viale delle Scienze, 90128 Palermo, Italy; stefano.beccari@unipa.it

**Featured Application: The specific application of this research could be the implementation of a hydrogen-fueled engine in a hybrid electric vehicle in order to improve the engine efficiency and to eliminate its exhaust emissions.**

**Abstract:** Hybrid electric vehicles are currently one of the most effective ways to increase the efficiency and reduce the pollutant emissions of internal combustion engines. Green hydrogen, produced with renewable energies, is an excellent alternative to fossil fuels in order to drastically reduce engine pollutant emissions. In this work, the author proposes the implementation of a hydrogen-fueled engine in a hybrid vehicle; the investigated hybrid powertrain is the power-split type in which the engine, two electric motor/generators and the drive shaft are coupled together by a planetary gear set; this arrangement allows the engine to operate independently from the wheels and, thus, to exploit the best efficiency operating points. A set of numeric simulations were performed in order to compare the gasoline-fueled engine with the hydrogen-fueled one in terms of the thermal efficiency and total energy consumed during a driving cycle. The simulation results show a mean engine efficiency increase of around 17% when fueled with hydrogen with respect to gasoline and an energy consumption reduction of around 15% in a driving cycle.

**Keywords:** hydrogen; hybrid electric vehicle; pollutant emissions

## 1. Introduction

Road vehicles, both heavy and light duty, are primarily responsible for the pollutant and greenhouse gas emissions of the transportation sector. In recent decades, great research efforts have been made to reduce the environmental impact of road vehicles following two main strategies: the implementation of non-fossil fuels (bio-fuels, green hydrogen and electricity from renewable sources) and overall vehicle efficiency increases. Hybrid Electric Vehicles (HEV) are an example of the second strategy since they implement an Internal Combustion Engine (ICE), which is the main energy source, coupled with one or more electric machines and a battery pack that are meant to support the ICE during transient operations and to recover the vehicle's kinetic energy during deceleration phases.

The power-split powertrain configuration is one of the most economic and efficient; it is endowed of a planetary gear set (the power split device) that couples together the ICE, two electric motor/generators and the driveshaft connected to the wheels. This particular arrangement allows to operate the engine independently from the wheels and then to exploit, for each road load condition, the best ICE efficiency operating condition [1,2]. As far as greenhouse gasses emission reductions are concerned, the most promising alternative to fossil fuels, as an energy carrier, is represented by the so-called green hydrogen, produced by water electrolysis with renewable energy sources [3,4]. A stoichiometric air-hydrogen mixture has a volumetric power density comparable with gaseous hydrocarbons, hydrogen can be stored and transported, and its combustion produces only water.

There is a huge amount of literature on the use of hydrogen as a fuel in ICE [5–8] exploring the pro and cons of its application; however, one undeniable conclusion is that the engine emissions are almost free from carbon dioxides (except the few coming from engine lubricant) and under fuel lean combustion operation, also free from nitrous monoxide [5].

Starting from the two important conclusions that hybridization improves the ICE engine efficiency and that the hydrogen-fueled ICE is the cleanest possible, the author explored the possibility to hybridize a hydrogen-fueled ICE in order to compare its efficiency with that of the same hybridized engine when fueled with gasoline; a series of numeric simulations, detailed in the subsequent section, was made with the purpose to compare the hydrogen-fueled ICE with the gasoline one in terms of the Brake Thermal Efficiency (BTE) and total energy consumption in a driving cycle (DC). The implementation of a hydrogen-fueled ICE in an HEV is not new, some examples can be found in the literature—both theoretical [9–11] and practical [12–14]—proving that the topic is attractive, and the present work fit perfectly into this trend.

In [11], a fuel consumption reduction of 12.6% was reported for a hydrogen-fueled engine, in an HEV, compared to its gasoline counterpart; in [12], a 14.32% reduction in DC fuel consumption was reported for a hydrogen-enriched diesel engine, in an HEV, with respect to the standard diesel engine equipped in a conventional vehicle. In [13], a 20% fuel consumption reduction was reported when comparing a hydrogen-fueled HEV with a gasoline-fueled conventional vehicle.

All the above-mentioned studies either refer to hydrogen-enriched conventional fuels or refer to existing conventional engines converted to hydrogen and implemented in HEV but are experimental studies; there are few pure theoretical studies proving the superiority of hydrogen over gasoline in an HEV based on engine fuel consumption maps as performed in this work. The results of the present theoretical study largely confirm the engine efficiency and fuel consumption improvements reported in the literature.

The work proposed here starts from an existing spark-ignition ICE whose behavior, in terms of performance efficiency and pollutant emissions, has been widely explored experimentally both in gasoline and in natural gas (NG) operating modes [15–21]. The engine has been simulated with a zero-dimensional (0-D) thermodynamic model [22,23] that, after proper calibration with experimental data [22], is able to predict the engine performance in terms of BTE or Brake Specific Fuel Consumption (BSFC) maps.

In the present work, a simulated BTE map of the mentioned engine fueled with hydrogen, as detailed in the following section, was obtained and compared with the BTE map of the gasoline-fueled engine in order to highlight the advantages of the hydrogen fuel in terms of engine BTE increases and DC energy consumption reduction. Three different comparisons were performed: at first the mean engine BTE, evaluated over all the possible engine operating conditions, was compared referring to a conventional vehicle application, then the mean engine BTE, evaluated only over the best efficiency operating points, was compared referring to an HEV that is able to, due to its power-split arrangement, exploit the ICE engine only in its best efficiency operating points.

Finally, the DC energy consumption of the hydrogen hybrid vehicle was compared with both the gasoline conventional and the gasoline hybrid vehicle. The results of all comparisons are in favor of the hydrogen-fueled engine, which proves to be the cleanest and most efficient candidate to substitute, in the near future, fossil fuels as an ICE power source in both conventional and hybrid vehicles.

## 2. Numerical Simulations

A 0-D thermodynamic model, described in [22], properly calibrated using experimental data coming from a spark-ignition engine fueled with both gasoline and gaseous fuels [22,23], was used in the present work to obtain BTE maps of the engine when fueled with both gasoline and hydrogen. Table 1 shows the ICE specifications.

**Table 1.** Internal Combustion Engine (ICE) specifications.

| Engine Specification | Value |
| --- | --- |
| brand and model | FIAT-FIRE 1.2 8v |
| $n$. of cylinders | 4 |
| Bore | 70.8 (mm) |
| Stroke | 78.86 (mm) |
| Rod to crank ratio | 3.27 |
| Compression ratio | 9.8 |
| Engine displacement | 1242 ($cm^3$) |
| Engine maximum power | 43.1 (kW) |

The main model inputs are the engine speed, engine load (Manifold Absolute Pressure or MAP), air/fuel mass ratio and spark advance (SA); the main model outputs are the engine indicated and brake mean effective pressures (IMEP and BMEP), the engine indicated and brake thermal efficiencies (ITE and BTE) and the engine torque and power. With respect to the previous simulation version [22], which was able to evaluate only the in-cylinder pressure during the whole engine cycle and then the IMEP, an engine friction model was added in order to evaluate the friction mean effective pressure (FMEP) and, in turn, both the BMEP and BTE. The Chen and Flynn friction model [24] was implemented; according to this model, the engine FMEP can be evaluated using the following equation:

$$FMEP = A + B \cdot p_{max} + C \cdot n + D \cdot n^2 \tag{1}$$

where $p_{max}$ is the maximum in-cylinder pressure; $n$ is the engine speed; and A, B, C and D are calibration coefficients that were tuned by means of experimental data in [15].

To identify the air/fuel (A/F) ratio of a mixture, the coefficient of excess air $\lambda$ is defined, it is the fraction between the actual A/F ratio and the stoichiometric one. In order to avoid dangerous combustion phenomena, such as pre-ignition and knocking, the hydrogen-fueled ICE must be operated with a lean A/F mixture [5–8] (i.e., with $\lambda > 1$), and this (together with the lower specific power of hydrogen) reduces the engine performance with respect to the gasoline operation mode; to fill this gap, one of the most used techniques is supercharging. In the literature, many combinations of supercharging pressure and $\lambda$ values can be found that lead to knock-safe operation and allow obtaining the same performance as gasoline-fueled engine [5,25–27].

The only pollutant emitted by a hydrogen-fueled engine, with a stoichiometric A/F ratio, is nitrous oxides (NOx); however, as soon as the A/F mixture is doubled (i.e., $\lambda = 2$) a drastic drop in NOx emissions is found [5,25,26]. Even supercharged engines do not emit NOx as long as the maximum combustion temperature remains below 1800 K [5], and in all the simulations presented in this work, the maximum combustion temperature remains below 1700 K.

Resuming the literature findings, one can roughly say that, with $\lambda = 2$ and a supercharging pressure of 2 bar absolute, the hydrogen-fueled engine is able to restore the gasoline engine performances and almost eliminate the NOx emissions. Regarding the combustion speed, hydrogen exhibits a laminar flame velocity one order of magnitude greater than gasoline for the stoichiometric A/F ratio and around 60% greater than gasoline for $\lambda = 2$ [27,28]. Considering the same engine and operating conditions, the turbulence inside the combustion chamber should be the same for both gasoline and hydrogen fuels, and then the ratio between the laminar flame velocities of the two fuels can be roughly considered the same between the turbulent flame velocities.

This conclusion was confirmed by some preliminary experimental tests performed by the author on an ICE test bench. Resuming all the above-mentioned considerations, in order to simulate a hydrogen-fueled engine that should produce the same performance of the gasoline counterpart (i.e., the maximum power), that should not produce abnormal combustion phenomena and that should not emit NOx, the combustion duration must be

set to roughly 60% of the gasoline counterpart, the A/F ratio must be set to $\lambda = 2$, and the engine must be supercharged in order to obtain MAP = 2 bar absolute.

The engine model was then upgraded in order to implement supercharging (SC) in two different ways: using a turbocharger (TC) driven by an exhaust gas turbine or using a volumetric compressor (VC) driven directly by the engine. In the first case, the engine MAP is set to a desired value that, for the sake of simplicity, was kept constant as the engine speed varied; the engine back pressure ($p_b$) was evaluated supposing that the same mass flows through both the compressor and turbine and equating the two specific works with the following equation:

$$p_b = \left( 1 - \frac{T_{MAN} \left( MAP^{\frac{k-1}{k}} - 1 \right)}{\eta_c \eta_t T_{EXMAN}} \right)^{\frac{-k}{k-1}} \tag{2}$$

where $k$ is the isentropic coefficient (1.4 for air); $T_{MAN}$ is the inlet manifold air temperature that is set to the ambient temperature (300 K) considering the presence of an intercooler; $\eta_c$ and $\eta_t$ are the compressor and turbine isentropic efficiencies set to 0.85 and 0.9, respectively; and $T_{EXMAN}$ is the exhaust manifold gas temperature set to a first-try value and then verified by simulations with an iterative process.

In the case of volumetric compressor supercharging (VCSC), as far as the compressor is driven directly by the engine, the corresponding FMEPc must be evaluated in order to obtain: BMEP = IMEP − FMEP − FMEPc. Assuming, for the sake of simplicity, an adiabatic screw compressor, the same volumetric efficiency for both engine and compressor and the presence of an intercooler between compressor and engine, one obtains:

$$FMEP_c = \frac{k}{k-1} MAP \left( MAP^{\frac{k-1}{k}} - 1 \right) \tag{3}$$

The desired MAP value depends on the compressor displacement $V_c$ and rotating speed $n_c$:

$$MAP = \frac{2 V_c n_c}{V n} \tag{4}$$

where $V$ and $n$ are the engine displacement and rotating speed, respectively.

Three engine configurations were simulated: the naturally aspirated gasoline-fueled engine, the turbocharged (TC) hydrogen-fueled engine and the supercharged (SC), with volumetric compressor (VC) hydrogen-fueled engine. For all the configurations, the engine speed and MAP were varied obtaining many different operating conditions. For each operating condition, $\lambda$ and SA were set as the optimal values (i.e., the maximum brake torque SA) or the knock-limited values (in the case of gasoline-fueled engines, those values were experimentally pre-determined [15–17]).

Some preliminary simulations were performed in order to find the MAP value of the boosted hydrogen-fueled engine able to produce the same maximum power of the gasoline version (43.1 kW): for the TC version, this value was MAP = 2.1 bar absolute (in accordance with the above-mentioned considerations) while, for the VCSC version, the value was MAP = 2.7 bar absolute; the second MAP value was higher because of the higher energy needed to drive the compressor.

Figure 1 shows the $\lambda$ and SA maps used in the gasoline-fueled engine simulation for all the operating conditions. The SA value is expressed in crank angle degrees (CAD) before top dead center (BTDC).

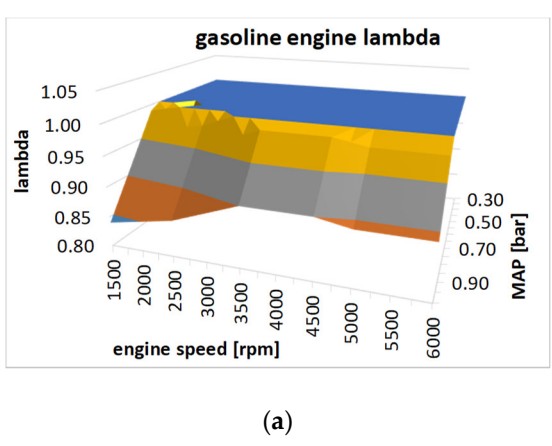 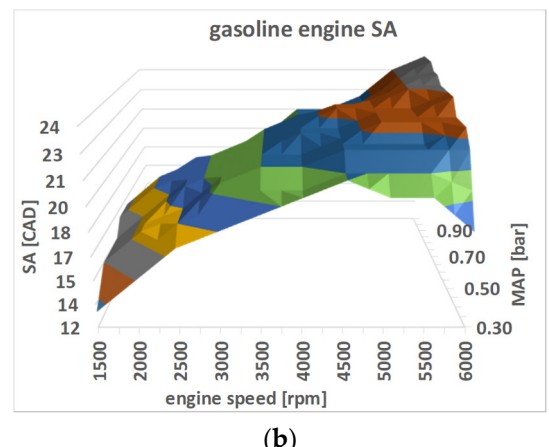

(**a**) (**b**)

**Figure 1.** λ (**a**) and SA (**b**) maps of the gasoline-fueled engine vs. engine speed [rpm] and MAP [bar].

For both hydrogen configurations, the A/F ratio is fixed at λ = 2 for all the operating conditions, while the SA is set to the maximum brake torque value. Figure 2a,b shows the SA maps adopted for the TC and SC engine, respectively.

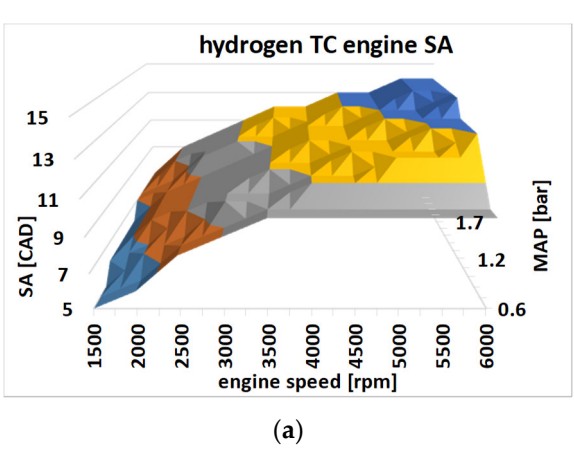 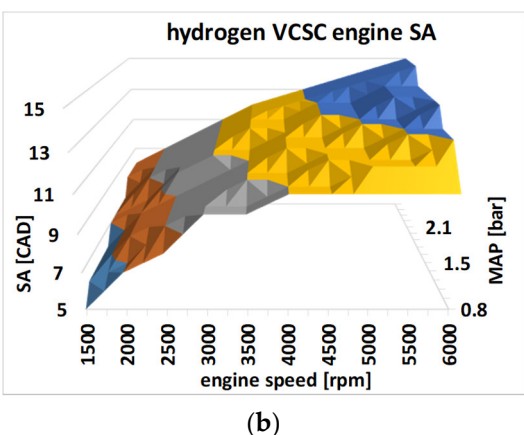

(**a**) (**b**)

**Figure 2.** SA maps of the TC (**a**) and VCSC (**b**) hydrogen-fueled engine vs. engine speed and MAP.

The main differences between gasoline- and hydrogen-fueled engine simulations are that the gasoline engine is operated with a rich A/F mixture and limited SA near full load conditions (Figure 1) to avoid knocking phenomena, and this lowers the engine BTE compared to hydrogen configurations; and the combustion duration of a gasoline-fueled engine is set to around 80 CAD [16,22,23] while the hydrogen combustion duration is set to around 50 CAD (roughly 60% lower than gasoline as stated above), and this further increases the hydrogen engine BTE compared to gasoline operation.

As already stated before, the hybridization obtains the best results for a conventional engine, and the scope of this work is to compare the hydrogen- and gasoline-fueled engines on their best operating conditions. The ICE is supposed to be implemented in a series-parallel hybrid vehicle equipped with a power-split device and two electric motor-generators in order to obtain the advantages of both series and parallel configurations as well as a high voltage battery pack (140 V) able to both recover the kinetic energy of the decelerating vehicle and drive the electric motor during transient operations.

The electric motors maximum power can be set to around 80% of the ICE maximum power (36 kW), and the battery pack energy capacity can be set to around 900 Wh. The above-described hybrid vehicle specifications correspond to that of a light duty passenger car [29].

In order to test the three hybridized engines (gasoline, VCSC and TC hydrogen) in a realistic situation, the Worldwide Harmonized Light Vehicles Test Cycle (WLTC) was considered. The cycle is composed of 1800 speed and acceleration values divided by a one second time interval (a fine discretization). For each speed value of the WLTC, the power required by the vehicle (PRV) was evaluated as the product between the speed and the force required to push the vehicle at that speed, sum of the acceleration force $F_a$ and the drag force $F_d$. Since both the speed and acceleration are known, for each operating point of the WLTC, it is possible to evaluate $F_a$ with the following equation:

$$F_a = (m + m_r)a \tag{5}$$

where $m$ is the vehicle curb weight, a is its acceleration and $m_r$ is the total equivalent mass of the rotating elements.

$F_d$ is the sum of aerodynamic and rolling resistance forces and can be evaluated with the following equation:

$$F_d = \frac{1}{2}A_d\rho v^2 + mgf_r \tag{6}$$

where $\rho$ is the air density, $v$ is the vehicle speed, $A_d$ is the drag area (product of the vehicle frontal area and the drag coefficient), $g$ is the gravitational acceleration and $f_r$ is the wheels' rolling friction coefficient. Assuming $m_r$ = 45 kg, $m$ = 1000 kg, $A_d$ = 0.75 m² (mean values of a standard passenger car [29]) and $f_r$ = 0.01, the PRV was evaluated for each point of the WLTC. The power supplied by the engine (PSE) was evaluated with the following equation:

$$PSE = \frac{PRV}{\eta_t} \tag{7}$$

where $\eta_t$ is the transmission efficiency for which a value of 0.9 can be set [30–33].

Assuming that in a split-power HEV the engine is always operated in its best efficiency points, for each WLTC operating point a specific procedure (described in the following section) was used to link a specific engine efficiency to each *PSE* and then to evaluate the power required by the engine (PRE); integrating the PRE values along the 1800 s of the WLTC duration, the total required energy was found.

Considering that an HEV is able to recover a part of the energy required to decelerate the vehicle, all the operating points of WLTC that involve a negative *PRV* were integrated, and the corresponding total energy, multiplied by an efficiency factor, was subtracted to the total required energy to obtain the total energy consumption. The efficiency factor, considering the conversion efficiency of the electrical machines and batteries involved, was set to a conservative value of 0.65.

To have a "low efficiency" reference, the conventional gasoline engine was tested with the above-described procedure; a five-speed gearbox with the following gear ratios was set: $t_1$ = 0.36, $t_2$ = 0.53, $t_3$ = 0.85, $t_4$ = 1.14 and $t_5$ = 1.22, and the final drive ratio was 0.29. For each WLTC vehicle speed, the engine speed was evaluated considering the following gear change speed for each gear ratio: $v_1$ = 35 km/h, $v_2$ = 70 km/h, $v_3$ = 112 km/h and $v_4$ = 168 km/h.

For each WLTC point, knowing the engine speed and the *PSE,* a specific engine BTE was taken from the simulated engine map in Figure 3a (shown in the following section) and used to evaluate the corresponding PRE; integrating all the PRE values over the WLTC time duration, the total required energy was evaluated and, due to the absence of energy recovery systems, coincides with the total energy consumption. Clearly, this procedure does not exploit the best engine efficiency points because the operating condition is bound by the engine speed that, in turn, is bound by the fixed engine-driveshaft gear ratio.

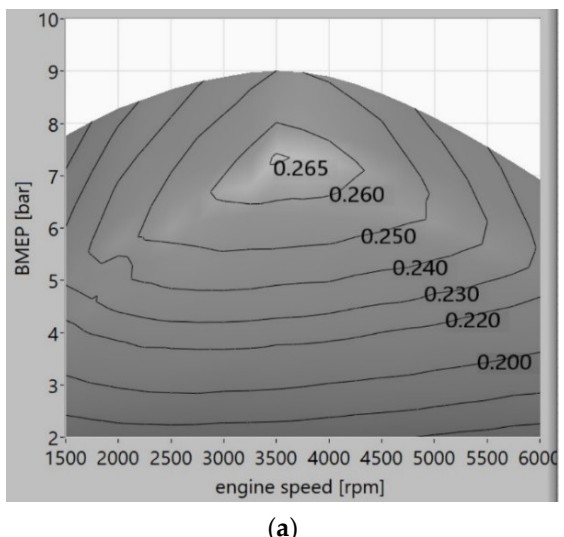 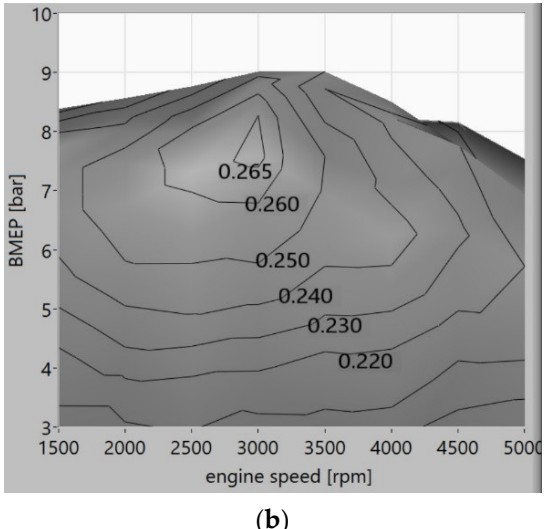

(**a**)  (**b**)

**Figure 3.** Simulated (**a**) and experimental (**b**) BTE maps of the gasoline-fueled engine.

## 3. Results and Discussion

The results of gasoline- and hydrogen-fueled engine simulations are presented in terms of BTE instead of BSFC because of the different lower heating values of the two fuels. Figure 3 shows the simulated (a) and experimental (b) BTE maps of the gasoline engine as a function of engine speed and BMEP (proportional to engine torque). The maximum efficiency is 0.265, and it is located below the full load condition for the above-mentioned considerations.

Figure 4 shows the BTE maps coming from the hydrogen-fueled engine simulation with the TC (a) configuration and the VCSC (b) configuration; some relevant aspects can be highlighted: the maximum BTE of the SC engine (0.277) is higher than the gasoline one but lower than the TC one (0.305); furthermore, the maximum BTE zone of the SC engine is well below the full load condition, such as in the gasoline BTE map while, in the TC configuration, the maximum efficiency zone is located at maximum BMEP. The TC configuration performs better than the VCSC one likely due to high compression work at high MAP levels in the second case.

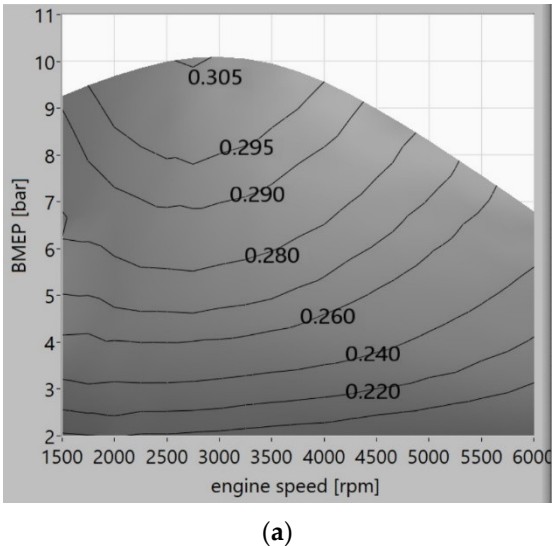 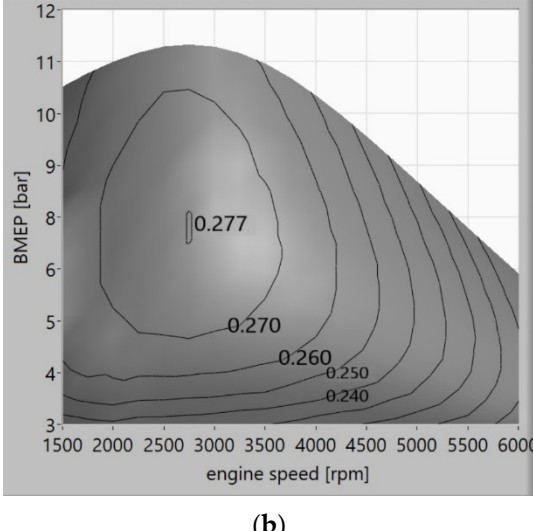

(**a**)  (**b**)

**Figure 4.** BTE maps of the hydrogen-fueled engine.

When the engine equips a conventional vehicle, for a given road load (i.e., a required power), there are infinite possible combinations of engine speed and load (operating points) that produce that power; however, only one is the best efficiency point. This scenario is represented in Figure 5 where the gasoline-fueled engine BTE map is displayed together with two curves at constant required power (10 and 20 kW); the curves cross the BTE surface in many different points but only the two marked with a green cross are the best efficiency points (i.e., the points where the constant power curve is tangent to a constant BTE curve).

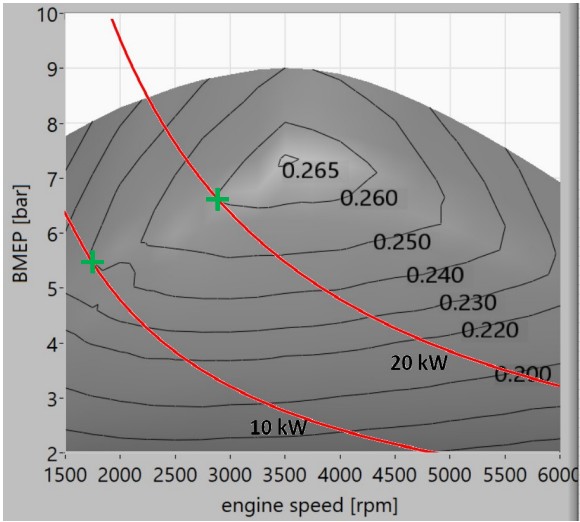

**Figure 5.** The best efficiency points at constant required power (gasoline).

The main difference between a conventional vehicle and a split-power hybrid one is that, in the first case, the ICE operating point depends on both the vehicle operating point and the current gear ratio between engine shaft and wheels while, in the second case, the engine can always be operated in the best efficiency points due to the decoupling between ICE and wheels and to the application of energy management control strategies (EMCS).

To find the mean BTE of the three engine configurations described above, equipping a conventional vehicle, one should evaluate the average BTE over all the points in the maps reported in Figures 3a and 4a,b because all the points are potential operating conditions.

Table 2 shows the mean BTE comparison for the three analyzed engine configurations: the same trend of the maximum BTEs is observed with the TC hydrogen-fueled engine that outperforms the gasoline operated one (almost 20% mean BTE increase), and the TC setup is slightly better than the VCSC one.

**Table 2.** The mean BTE of the three engine configurations in a conventional vehicle.

| Engine Configuration | Mean BTE | % Increase Compared to Gasoline |
|---|---|---|
| gasoline | 0.211 | 0 |
| hydrogen VCSC | 0.246 | 16.5% |
| hydrogen TC | 0.253 | 19.5% |

As already stated before, the hybridization obtains the best results from a conventional engine, and the scope of this work is to compare the hydrogen- and gasoline-fueled engines at their best operating conditions. Assuming that, theoretically, a split-power HEV is able to operate the engine in its higher BTE operating point for each required power, the best efficiency path (BEP) is defined as the curve in the BTE map that links all the maximum BTE points for each possible required power from zero to the maximum (43.1 kW).

Figure 6 shows the gasoline BTE map with different constant power curves and, in green, the BEP that links all the tangent points between a constant power curve and an iso-efficiency curve. The mean engine BTE, in this case, was evaluated as the average value over all the points lying in the BEP.

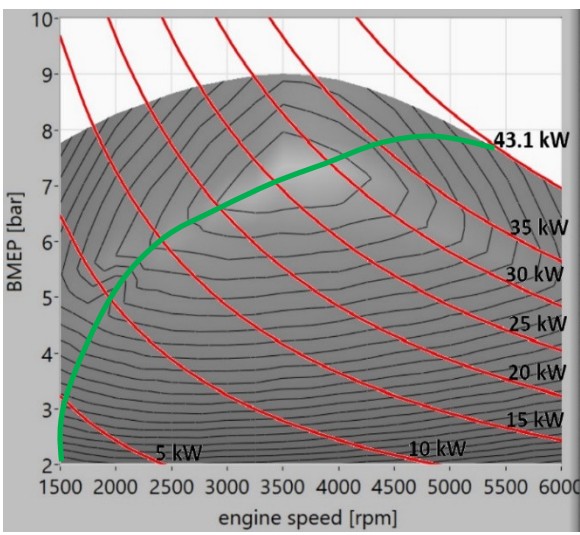

**Figure 6.** The gasoline-engine BTE map with the BEP in green.

Figure 7 shows the BTE maps of hydrogen-fueled engines in both TC and VCSC configuration with the BEP curves. The mean BTE value of the hydrogen-fueled engine equipped in an HEV is evaluated by averaging the BTE values lying on the BEP curves.

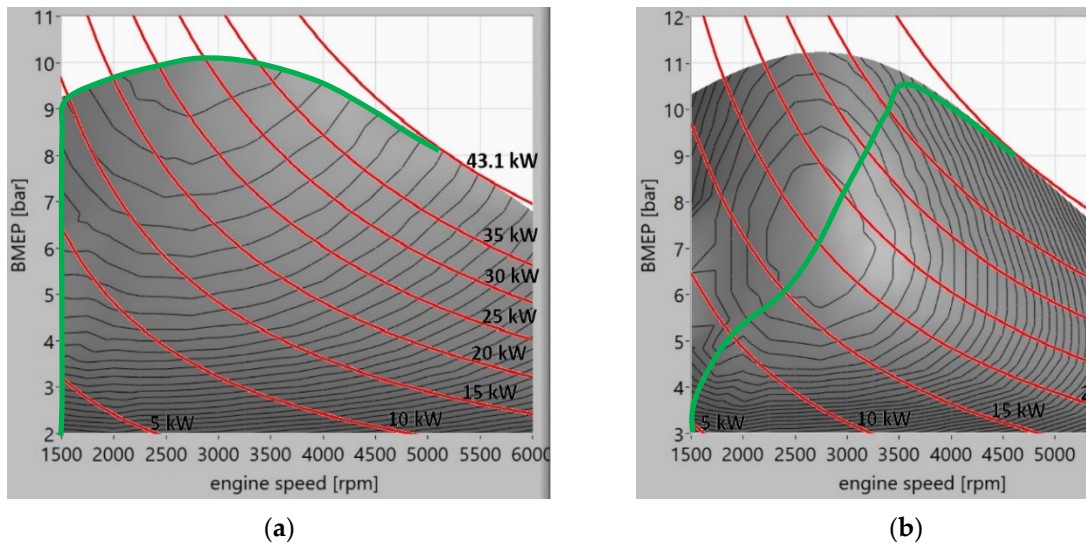

(**a**)                                                  (**b**)

**Figure 7. The** hydrogen-engine BTE map with the BEP in green.

Table 3 shows, for all the three engine configurations, the power and BTE values of all the points lying on the BEP.

**Table 3.** The engine power and BTE of points lying on BEPs.

| Power [kW] Gasoline | BTE Gasoline | Power [kW] SC Hydrogen | BTE SC Hydrogen | Power [kW] TC Hydrogen | BTE TC Hydrogen |
|---|---|---|---|---|---|
| 3.9 | 0.182 | 4.2 | 0.226 | 3.6 | 0.212 |
| 4.6 | 0.195 | 5.4 | 0.247 | 4.5 | 0.232 |
| 5.4 | 0.206 | 6.4 | 0.258 | 5.5 | 0.248 |
| 6.9 | 0.223 | 7.4 | 0.263 | 6.3 | 0.259 |
| 7.6 | 0.230 | 8.3 | 0.266 | 7.2 | 0.266 |
| 8.4 | 0.233 | 9.2 | 0.267 | 8.0 | 0.271 |
| 9.2 | 0.236 | 10.1 | 0.268 | 8.8 | 0.276 |
| 10.1 | 0.241 | 10.9 | 0.268 | 9.6 | 0.280 |
| 11.0 | 0.238 | 11.9 | 0.268 | 10.3 | 0.279 |
| 11.9 | 0.247 | 12.7 | 0.272 | 11.1 | 0.282 |
| 12.4 | 0.245 | 13.9 | 0.272 | 11.9 | 0.285 |
| 13.6 | 0.251 | 14.5 | 0.274 | 12.7 | 0.287 |
| 14.9 | 0.249 | 15.9 | 0.275 | 13.6 | 0.289 |
| 15.5 | 0.253 | 16.4 | 0.276 | 14.4 | 0.291 |
| 16.9 | 0.253 | 17.2 | 0.275 | 15.2 | 0.292 |
| 17.4 | 0.255 | 17.9 | 0.276 | 16.2 | 0.294 |
| 18.9 | 0.258 | 19.4 | 0.276 | 17.2 | 0.296 |
| 19.2 | 0.257 | 20.9 | 0.276 | 18.9 | 0.298 |
| 20.9 | 0.262 | 21.6 | 0.277 | 20.0 | 0.300 |
| 22.0 | 0.250 | 21.8 | 0.276 | 21.6 | 0.301 |
| 22.8 | 0.263 | 23.6 | 0.276 | 22.9 | 0.302 |
| 24.1 | 0.257 | 24.8 | 0.275 | 24.4 | 0.302 |
| 25.5 | 0.255 | 25.3 | 0.275 | 25.8 | 0.304 |
| 26.6 | 0.265 | 26.3 | 0.274 | 27.0 | 0.303 |
| 27.8 | 0.261 | 27.0 | 0.274 | 28.7 | 0.305 |
| 28.4 | 0.264 | 28.7 | 0.273 | 29.6 | 0.302 |
| 29.1 | 0.258 | 29.1 | 0.272 | 31.3 | 0.304 |
| 30.1 | 0.256 | 30.3 | 0.271 | 31.9 | 0.301 |
| 31.4 | 0.261 | 31.8 | 0.269 | 33.8 | 0.303 |
| 32.6 | 0.258 | 32.6 | 0.268 | 34.1 | 0.300 |
| 33.8 | 0.257 | 33.3 | 0.267 | 36.0 | 0.302 |
| 34.6 | 0.254 | 34.4 | 0.266 | 37.4 | 0.294 |
| 36.1 | 0.252 | 35.7 | 0.265 | 38.0 | 0.299 |
| 37.5 | 0.250 | 36.1 | 0.264 | 38.6 | 0.290 |
| 38.5 | 0.247 | 37.7 | 0.261 | 39.6 | 0.296 |
| 39.9 | 0.245 | 39.2 | 0.259 | 40.8 | 0.292 |
| 41.0 | 0.241 | 40.8 | 0.254 | 41.8 | 0.288 |
| 41.9 | 0.237 | 42.0 | 0.249 | 42.5 | 0.283 |
| 42.6 | 0.234 | 42.7 | 0.243 | 42.9 | 0.277 |
| 43.1 | 0.226 | 43.1 | 0.236 | 43.1 | 0.271 |

Table 4 shows the average values extracted from Table 3, which are the mean BTE of a gasoline- or hydrogen-fueled ICE when equipped in an HEV. Some important considerations can be drawn: The ICE hybridization brings great benefits to the mean BTE in both gasoline (+15.9%) and hydrogen (+8.0% and +13.3%) configurations. TC remains, also in HEV, a better alternative than VCSC. Finally, the TC hydrogen ICE has a higher mean BTE than the gasoline hybridized engine in both the conventional and hybridized configurations with increases of +3.1% and +16.7%, respectively.

**Table 4.** The mean BTE of the three engine configurations in an HEV.

| HEV ICE Configuration | Mean BTE | % Increase Compared to Gasoline Hybrid | % Increase Compared to Conventional |
|---|---|---|---|
| gasoline | 0.245 | 0 | 15.9% |
| hydrogen VCSC | 0.266 | 8.6% | 8.0% |
| hydrogen TC | 0.286 | 16.7% | 13.3% |

In order to test the three hybridized engines (gasoline, VCSC and TC hydrogen) in a more realistic situation, they were implemented in the WLTC driving cycle as described above. Table 5 resumes the findings, in terms of the total energy consumed in the WLTC, of both gasoline- and hydrogen-fueled engines. All the hybrid configurations show a strong reduction of energy consumption compared to the conventional gasoline engine, from the −15% of the hybrid gasoline to the −23.6% of the VCSC hybrid and finally to the −28.2% of the TC hybrid, and this confirms the effectiveness of the hybrid technology in increasing the vehicle efficiency by exploiting the engine best efficiency operating points and by recovering the vehicle kinetic energy.

Table 5 highlights the efficiency increases specifically due to the use of hydrogen in a hybrid vehicle with respect to gasoline (third column). The hydrogen VCSC hybrid engine shows a total energy consumption that is 9.7% lower than the gasoline hybrid, and the hydrogen TC hybrid engine is 15.1% lower than the gasoline hybrid confirming the already discussed mean efficiency increases reported in Table 4.

The total fuel mass consumption of hydrogen (in both TC and VCSC configurations) was lower than one third that of gasoline, and the main reason is the lower heating value of hydrogen that is three times that of gasoline. In any case, a vehicle mileage evaluation can be made. The energy consumption reduction of 15.1% agrees well with the above-mentioned literature results [9–14] in which a range between 12% and 20% of gasoline equivalent fuel consumption reduction has been reported between conventional fuel and hydrogen in HEV.

**Table 5.** WLTC energy and fuel mass consumption comparison.

| ICE Configuration | Total Energy Consumption [kJ] | Energy Difference Compared to Gasoline Hybrid [%] | Energy Difference Compared to Gasoline Conventional [%] | Total Fuel Mass Consumption [kg] |
|---|---|---|---|---|
| gasoline conventional | 50,731 | +18.0% | 0 | 1.153 |
| gasoline hybrid | 42,930 | 0 | −15.0% | 0.976 |
| hydrogen VCSC hybrid | 38,749 | −9.7% | −23.6% | 0.323 |
| hydrogen TC hybrid | 36,430 | −15.1% | −28.2% | 0.304 |

Considering that a 50 L cylinder at 300 bar pressure contains 1.35 kg of hydrogen, and with 0.304 kg of hydrogen, a TC HEV can theoretically travel 23.262 km (WLTC equivalent travel distance), three cylinders would allow a 310.3 km travel distance, which is good mileage if compared with natural-gas-fueled light duty vehicles for which commercial data [34,35] and the scientific literature [36,37] report an average mileage of around 400 km considering a typical 13–14 kg fuel tank capacity.

To resume the findings, the hydrogen-fueled engine, also in its conventional configuration, performed better than the gasoline engine even if the latter was hybridized (Tables 2 and 4). If the hydrogen engine underwent hybridization, this performance gap increased even more (Table 4); this efficiency gap remained almost unchanged when analyzing the WLTC total energy consumption (Table 5), and a respectable mileage of more than 300 km was estimated for the TC hydrogen-fueled engine. This conclusion, together

with the fact that the hydrogen-fueled engine is almost free from pollutant and greenhouse gasses emissions proves the superiority of green hydrogen over traditional fossil fuels in both conventional and hybrid vehicles.

## 4. Conclusions

This paper shows a theoretical comparison between a gasoline- and a hydrogen-fueled ICE equipped in an HEV; a set of simulations was performed in order to compare the two fuels in terms of both the engine efficiency and total energy consumption while keeping the maximum engine power constant. To obtain the same maximum power of the gasoline-fueled engine, the hydrogen engine must be supercharged, and the best solution proved to be a turbocharger driven by an exhaust gas turbine. The first comparison involves a conventional vehicle: the hydrogen turbocharged engine exhibited an average BTE 19.5% higher than that of the gasoline counterpart.

A second comparison between the two fuels was performed considering the engine aboard an HEV that exploits, for each operating condition, the best engine efficiency. In this case, the hydrogen TC engine showed an average BTE 16.7% higher than that of the gasoline one. Finally, the WLTC driving cycle was considered in order to evaluate the total energy consumption of the engine equipped in the HEV and fueled either by gasoline or by hydrogen. In this case, the total energy consumption of the hydrogen-fueled engine (HEV) was 15.1% lower than that of the gasoline one (HEV) and 28.2% lower if compared with the gasoline engine aboard a conventional vehicle.

An indicative mileage of 300 km was estimated for the hybrid hydrogen engine with a 150 L tank loaded at 300 bar pressure. The energy consumption reduction of 15.1% agrees with the above-mentioned literature results in which a range between 12% and 20% of gasoline equivalent fuel consumption reduction was reported between conventional fuel and hydrogen in HEVs.

The higher engine efficiency when fueled with hydrogen compared with gasoline is mainly due to three reasons: the lean mixture composition (while gasoline, at full load and works with rich mixture), the better combustion phasing allowed by the higher knocking resistance and the shorter combustion duration due to the higher flame speed. In this work, the three mentioned advantages of hydrogen were quantified in terms of attainable engine efficiency increase and energy consumption reduction in HEVs.

Our conclusion is that a hydrogen-fueled engine performs better than the gasoline counterpart in both conventional and hybrid vehicles; considering that hydrogen, under proper operating conditions, is a zero-emissions fuel, this represents an almost obligatory choice in the automotive field.

**Funding:** This research received no external funding.

**Institutional Review Board Statement:** Not applicable.

**Informed Consent Statement:** Not applicable.

**Data Availability Statement:** Not applicable.

**Conflicts of Interest:** The authors declare no conflict of interest.

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
