# Peer review of "On the Use of a Hydrogen-Fueled Engine in a Hybrid Electric Vehicle"

_applsci, doi:10.3390/app122412749_

Round 1
Reviewer 1 Report
The paper "On the use of a hydrogen fueled engine in a hybrid electric vehicle" submitted for review is devoted to the possible use of a gasoline engine to run a hybrid car when converted to hydrogen fuel. Undoubtedly, the article arouses a certain interest among power engineering specialists. Particularly interesting are the results of the work in terms of predicting the energy possibilities of using hydrogen fuel in a car.
However, according to the content of the article, there are certain shortcomings for its improvement. This requires some clarifications to unequivocally improve the understanding of the material presented in the article:
1. It is difficult to understand what type of hybrid vehicle the engine was identified and matched for: micro hybrid, mild hybrid, full hybrid, plug-in hybrid (PHEV), parallel, series? It is not clear how the mode of operation of this hybrid is reflected in the text of the article - based on the features of the approach and calculation. The work, of course, lacks justification for the use of the engine for the intended purpose of the vehicle. Also, it would be useful to substantiate the characteristics of the drive motor under study and give a link to its technical documentation so that the reader can unambiguously determine the possibilities of its operational use in operating conditions.
2. Three engine configurations were simulated in the article: naturally aspirated gasoline engine, turbocharged hydrogen engine (TC), and supercharged hydrogen engine (SC) with positive displacement compressor (VC). However, the text of the article does not substantiate the option of using the engine, specifically for the purpose of using it to perform transport work, namely as a vehicle, and not as a stationary engine (for a power plant).
3. The statement that in the work after proper calibration with experimental data [22] it is possible to predict the characteristics of the engine in these options is very doubtful. Experimental studies for the engine described in the article in the configuration of a turbocharged hydrogen engine (TC) and a supercharged hydrogen engine (SC) with a positive displacement compressor (VC) are not given at all. In addition, when using references in the article to the results of experimental data that are not shown in terms of comparability / adequacy.
4. The problem considered in the article is presented for steady-state engine operation modes, i.e. in constant power modes. However, the option of working in transient modes is not considered. This is especially true for the operation of the vehicle (engine) in driving cycles.
5. The article does not describe the design changes made by the author in the engine to obtain a lean mixture (especially in the gasoline version). In supercharged versions of hydrogen, constructive solutions are also interesting for their implementation after the conversion of the engine to run on hydrogen.
6. It is not clear from the text of the article how the reconstruction of the engine and the reorganization of the engine workflow to a lean mixture was implemented in the mathematical model with which the studies were carried out.
7. It is not clear for which vehicle the studies were carried out, the characteristics of the vehicle are not clear. Even for the energy assessment of the fuel, it is necessary to show in the article the features of the vehicle under study.
8. The wording “Considering that a 50 liters cylinder at 300 bar pressure contains 1.35 kg of hydrogen and with 0.304 kg of hydrogen an HEV can theoretically travel 23.262 km (WLTC equivalent travel distance), two cylinders would allow a 206.9 km travel distance that is a quite good mileage for a gas fueled vehicle." is not correct enough, because comparison with a gas car was not carried out in this article (The study concerned only three types of engines: “Three engine configurations have been simulated: the naturally aspirated gasoline fueled engine, the turbocharged (TC) hydrogen fueled engine and the supercharged (SC), with volumetric compressor , (VC) hydrogen fueled engine."). It is necessary to make corrections to the article to clarify the information provided.
9. Definitely, the conclusions on the article are trivial and do not reflect the results of the study and the materials presented in the article ... The conclusions in the article need to be unambiguously corrected in the direction of the text of the article.
10. I believe that it would be quite interesting and useful to show the complete results obtained by the author in the implementation of the driving cycles of a hybrid vehicle for all three engine options (“Three engine configurations have been simulated: the naturally aspirated gasoline fueled engine, the turbocharged (TC) hydrogen fueled engine and the supercharged (SC), with volumetric compressor, (VC) hydrogen fueled engine."). This would make it possible to unambiguously show the dynamics of a hybrid vehicle, fuel efficiency and environmental friendliness of each vehicle engine. However, this needs to be confirmed by experimental data in order to conduct a computational-theoretical study.
Author Response
In the manuscript all the text that has been added is written in red.
reviewer:
- It is difficult to understand what type of hybrid vehicle the engine was identified and matched for: micro hybrid, mild hybrid, full hybrid, plug-in hybrid (PHEV), parallel, series? It is not clear how the mode of operation of this hybrid is reflected in the text of the article - based on the features of the approach and calculation. The work, of course, lacks justification for the use of the engine for the intended purpose of the vehicle. Also, it would be useful to substantiate the characteristics of the drive motor under study and give a link to its technical documentation so that the reader can unambiguously determine the possibilities of its operational use in operating conditions.
authors reply:
The hybrid powertrain is of the series-parallel power-split type; all the powertrain specifications have been added to the text; the ICE BTE experimental map has been added to the text; the hybridization allows to both operate the ICE in its best efficiency points for each operating conditions (the best efficiency path is introduced) and to recover the vehicle deceleration energy and improve the energy consumption during the driving cycle.
reviewer:
2. Three engine configurations were simulated in the article: naturally aspirated gasoline engine, turbocharged hydrogen engine (TC), and supercharged hydrogen engine (SC) with positive displacement compressor (VC). However, the text of the article does not substantiate the option of using the engine, specifically for the purpose of using it to perform transport work, namely as a vehicle, and not as a stationary engine (for a power plant).
authors reply:
In all the text of the article the engine is referred to work either in a traditional or in a hybrid vehicle (a light duty passenger car), it is expressely declared with clear sentences. In fact the driving cycle is taken in to consideration and the energy consumption is evaluated.
reviewer:
3. The statement that in the work after proper calibration with experimental data [22] it is possible to predict the characteristics of the engine in these options is very doubtful. Experimental studies for the engine described in the article in the configuration of a turbocharged hydrogen engine (TC) and a supercharged hydrogen engine (SC) with a positive displacement compressor (VC) are not given at all. In addition, when using references in the article to the results of experimental data that are not shown in terms of comparability / adequacy.
authors reply:
a lot of references reported in the manuscript refers to experimental tests, performed by the same author on the real engine object of the present study, when fueled with gasoline, natural gas and propane both in the naturally aspirated and in the supercharged configuration with volumetric compressor; this huge amount of experimental data allowed to accurately calibrate the 0D numerical model used in this work in many different operating conditions and to predict the gasoline engine BTE map. The hydrogen fueled engine maps have been obtained from the calibrated model making some hypothesis (the shorter combustion duration for example) that must be indeed experimentally validated (and this represent the future developments of this work) but the author believes that the results of the comparison between gasoline and hydrogen hybrid engines will not be greatly affected and remain valid.
reviewer:
4. The problem considered in the article is presented for steady-state engine operation modes, i.e. in constant power modes. However, the option of working in transient modes is not considered. This is especially true for the operation of the vehicle (engine) in driving cycles.
authors reply:
In the first part of the paper the operating conditions lying on the best efficiency path of the engine maps have been considered and the average efficiency have been evaluated considering a sum of steady-state conditions but in the second part of the work, when evaluating the energy consumption during a driving cycle, both the vehicle speed and acceleration have been considerd, for each pont of the driving cycle, to evaluate the power absorbed by the vehicle in terms of aerodynamic and rolling resistance and also in terms of inertia forces; moreover the points of the driving cycle are sampled with a one second time step that means a very fine discretization and this means, in turn, that all the transient phases have been represented by a great amount of successive steady-state conditions and this brings to almost identical results. In any case to study the specific engine behaviour in transient operation goes behind the scope of this work that focuses on a first approximation approach.
reviewer:
5. The article does not describe the design changes made by the author in the engine to obtain a lean mixture (especially in the gasoline version). In supercharged versions of hydrogen, constructive solutions are also interesting for their implementation after the conversion of the engine to run on hydrogen.
authors reply:
in gasoline mode, the engine is never fueled with lean mixture but in some operating conditions with rich mixture that is simply regulated, in the real engine, by electro-injectors. In the simulation the parameter that accounts for A/F ratio is the excess air rato "Lambda" that is the ratio between actual A/F ratio and stoichiometric A/F ratio. No design changes are needed to run the engine with different A/F ratios but simply force the injectors to properly change the injection time and this task is performed by the electronic control unit; in the simulation a proper "Lambda" map is loaded that forces a specific A/F ratio value for each operating condition.
In order to run the actual supercharged engine with hydrogen a proper injector must be equipped that is able to deliver the proper amount of gas to the relative cylinder of the engine; this is a well estabilished technology but it has nothing to share with the simulation of the engine and it is behind the scope of this work.
reviewer:
6. It is not clear from the text of the article how the reconstruction of the engine and the reorganization of the engine workflow to a lean mixture was implemented in the mathematical model with which the studies were carried out.
authors reply:
as already pointed out the mixture A/F ratio is a simple number that is changed in engine simulation and the real world conditions and equipments needed to run the engine with hydrogen (lean or stoichiometric) are not an authors concern because that is behind the scope of the present work and those conditions and equipments do not influence the engine simulation. The only influence that vehicle equipments have in this study is in terms of vehicle mass and this has been taken into account properly.
reviewer:
7. It is not clear for which vehicle the studies were carried out, the characteristics of the vehicle are not clear. Even for the energy assessment of the fuel, it is necessary to show in the article the features of the vehicle under study.
authors reply:
A reference to the vehicle specifications has been added to the text of the manuscript.
reviewer:
8. The wording “Considering that a 50 liters cylinder at 300 bar pressure contains 1.35 kg of hydrogen and with 0.304 kg of hydrogen an HEV can theoretically travel 23.262 km (WLTC equivalent travel distance), two cylinders would allow a 206.9 km travel distance that is a quite good mileage for a gas fueled vehicle." is not correct enough, because comparison with a gas car was not carried out in this article (The study concerned only three types of engines: “Three engine configurations have been simulated: the naturally aspirated gasoline fueled engine, the turbocharged (TC) hydrogen fueled engine and the supercharged (SC), with volumetric compressor , (VC) hydrogen fueled engine."). It is necessary to make corrections to the article to clarify the information provided.
authors reply:
a proper comparison with a natural gas vehicle has been added in the text through literature references and now the sentences regarding the hybrid hydrogen vehicles mileage are more consistent.
reviewer:
- Definitely, the conclusions on the article are trivial and do not reflect the results of the study and the materials presented in the article ... The conclusions in the article need to be unambiguously corrected in the direction of the text of the article.
authors reply:
the results section has been modified in the manuscript and now it reflects the findings of the study.
reviewer:
10. I believe that it would be quite interesting and useful to show the complete results obtained by the author in the implementation of the driving cycles of a hybrid vehicle for all three engine options (“Three engine configurations have been simulated: the naturally aspirated gasoline fueled engine, the turbocharged (TC) hydrogen fueled engine and the supercharged (SC), with volumetric compressor, (VC) hydrogen fueled engine."). This would make it possible to unambiguously show the dynamics of a hybrid vehicle, fuel efficiency and environmental friendliness of each vehicle engine. However, this needs to be confirmed by experimental data in order to conduct a computational-theoretical study.
authors reply:
the results of the energy comsumption section have been expanded in the manuscript and now the SCVC hydrogen engine is included but also, as reference, the traditional vehicle with gasoline engine consumption in order to have a more comprehensive view of the benefits obtainable by both the fuel and the hybridization. Undoubtedly those results must be experimentally validated and this will be the future development of the present study.
Reviewer 2 Report
Major Comments:
Section Results and discussion:
As you mentioned in Introduction, hydrogen hybrid powertrain system is not a new topic, and a lot of studies have been done on it. But you just illustrated your own results without comparing them with the results from existing studies, while I think this comparison is necessary. Only the comparison could reveal what’s new in your research and the contribution of your study.
Minor Comments:
Check the figure and table number in the text because there is some error.
Too many very large tables in this manuscript; maybe you can simplify the tables or illustrate them in other ways.
Is Table 7 essential here in your manuscript?
Enrich your conclusion because it seems too short.
Author Response
reviewer:
Section Results and discussion:
As you mentioned in Introduction, hydrogen hybrid powertrain system is not a new topic, and a lot of studies have been done on it. But you just illustrated your own results without comparing them with the results from existing studies, while I think this comparison is necessary. Only the comparison could reveal what’s new in your research and the contribution of your study.
authors reply:
The results of existing studies have been added in the manuscript and now a proper comparison with the results presented in this work can be made.
reviewer:
Check the figure and table number in the text because there is some error.
authors reply:
the errors have been corrected.
reviewer:
Too many very large tables in this manuscript; maybe you can simplify the tables or illustrate them in other ways. Is Table 7 essential here in your manuscript?
authors reply:
the tables reporting the engine A/F ratio and spark advance have been substituted with map diagrams and so the manuscript readability has been improved.
reviewer:
Enrich your conclusion because it seems too short.
authors reply:
the conclusion section has been enriched and the results better highlighted.
Reviewer 3 Report
1- Why lambda = 2 A/F ratio for hydrogen is adopted?
2- The engine speed-MAP tables 2-5 are not helpful to interpret any trend.
3- There is “Error! Reference source not found” in the text that must be revised by the authors before submission.
4- The reason why gasoline vs hydrogen engine is tested in HEV power mode has not been explained. What would be the difference if the comparison is made in the conventional SI engine?
5- The validity study of the applied 0D model has not proven here. I think it is demanding job to verify the results of different fuels with the equations.
6- How about the HEV specification? What effect battery/FC can have on the obtained results?
7- In Table 8, why BTE increase of hybrid is less than the base engine in TC case?
8- Neither abstract, nor conclusions do not summarize the findings.
Author Response
reviewer:
1- Why lambda = 2 A/F ratio for hydrogen is adopted?
authors reply:
for one hand in order to avoid nomalous combustion phenomena such as auto-ignition and knocking and on the other hand to eliminate NOx emissions, this has been extensively explained in the manuscript.
reviewer:
2- The engine speed-MAP tables 2-5 are not helpful to interpret any trend.
authors reply:
those tables have been substituted with map diagrams just to give the same information in a more effective way.
reviewer:
3- There is “Error! Reference source not found” in the text that must be revised by the authors before submission.
authors reply:
this error has been corrected.
reviewer:
4- The reason why gasoline vs hydrogen engine is tested in HEV power mode has not been explained. What would be the difference if the comparison is made in the conventional SI engine?
authors reply:
the comparison between gasoline and hydrogen engine has been performed both in conventional and in HEV power mode: the results clearly show that the hydrogen engine is better than the gasoline one in both power modes but since hybridization produces an increase of engine efficiency (both gasoline and hydrogen) it is very useful to ibridize the hydrogen engine because, in this way the two efficiency increments are explioted together: the increase due to the use of hydrogen instead of gasoline and the further increase due to hybridization. This is clearly explained in the manuscript.
reviewer:
5- The validity study of the applied 0D model has not proven here. I think it is demanding job to verify the results of different fuels with the equations.
authors reply:
the presented 0D model has been widely experimentally validated by the author with natural gas both in naturally aspirated and in supercharged mode. A further validation with hydrogen is needed, and will be the object of a future development of this work, in order to prove the validity of the hypothesys made in the present study that are however indeed supported by a wide literature.
reviewer:
6- How about the HEV specification? What effect battery/FC can have on the obtained results?
authors reply:
the HEV specifications have been added to the manuscript; the battery pack could influence the amount of energy stored and effectively given back to the vehicle during the driving cycle and in turn slightly influence the total amount of consumed energy and finally the vehycle mileage.
reviewer:
7- In Table 8, why BTE increase of hybrid is less than the base engine in TC case?
authors reply:
in former Table 8 (now Table 4) two different comparisons have been made: the hybridized engine (each configuration) has been compared with its conventional counterpart (same fuel and configuration) to highlight the positive effect of hybridization (4th column in Table 4), in this case the TC hydrogen increases its BTE of 13.3% with respect to TC hydrogen conventional; another comparison regards the benefits of hydrogen fuel compared to gasoline in a HEV and the 3th column in Table 4 reports that TC hydrogen hybrid exhibits a 16.7% BTE increase compared to gasoline hybrid. In Table 4 hence the two different benefits regarding either hybridization or fuel type are highlighted and the percentage inctements are not related to each other so no point in evidencing that one is higher of the other or vice versa. If the reviewer instead is asking why in 4th column of Table 4 the BTE increment of TC hydrogen with respect to conventional is lower than in the gasoline case this is due to the high average BTE of hydrogen conventional engine that mitigates the positive effect of hybridization; on the contrary, the gasoline conventional engine low average BTE amplifies the hybridization positive effect. In any case is widely proved that both hybridization and hydrogen bring positive effects on engine BTE so the best chioce is to implement both of them.
reviewer:
8- Neither abstract, nor conclusions do not summarize the findings.
authors reply:
the manuscript has been modified and now both abstract and conclusions summarize the findings.
Reviewer 4 Report
The manuscript reports the “On the use of a hydrogen fueled engine in a hybrid electric vehicle”. The authors have provided significant new understanding in this manuscript. The results have been explained reasonably well. Although the basis of the articles in interesting I think that some clarifications are needed in order to clarify the real novelty of the work.
· Comments 1: Please improve abstract. The abstract (about 150 words) should be informative, concisely stating the subject, and giving a clear indication of the nature and range of the results contained in the paper. Abstracts, titles and keywords draw more attention of the potential readers/researchers and increase discoverability of the papers. Rewrite the abstract.
· Comments 2: Check the space and typological error throughout the manuscript.
· Comments 3: Authors cite recent references like 2020, 2021 and 2022.
· Comments 4: Check the unit’s mm as mM, minutes as min, Seconds as S, hours as h, ml as mL and also check the space and typological error throughout the manuscript.
· Comments 5: Authors must check the journal format.
· Comments 6: Authors must concentrate the clarity of the figures.
· Comments 7: Authors add five keywords in the manuscript.
· Comments 8: Authors must add the following references
o https://doi.org/10.1016/j.ijhydene.2019.03.131
o https://doi.org/10.1016/j.fuel.2020.118120
o https://doi.org/10.1016/j.fuel.2020.117645
o https://doi.org/10.1007/s13204-021-01781-z
o https://doi.org/10.1016/j.scitotenv.2020.142389
· Comments 9: Authors add the line numbers and page numbers.
· Comments 10: Authors must add some topics in the section 2.
· Comments 11: Authors must check the grammar throughout the manuscript.
Author Response
reviewer:
Comments 1: Please improve abstract. The abstract (about 150 words) should be informative, concisely stating the subject, and giving a clear indication of the nature and range of the results contained in the paper. Abstracts, titles and keywords draw more attention of the potential readers/researchers and increase discoverability of the papers. Rewrite the abstract.
authors reply:
the abstract has been modified to better highlight the findings.
reviewer:
Comments 2: Check the space and typological error throughout the manuscript.
Comments 4: Check the unit’s mm as mM, minutes as min, Seconds as S, hours as h, ml as mL and also check the space and typological error throughout the manuscript.
authors reply:
the manuscript has been checked.
reviewer:
Comments 3: Authors cite recent references like 2020, 2021 and 2022.
authors reply:
in the manuscript both recent and old references have been cited and the choice is not based on the publication year but rather on the reference content and validity. I don't understand the reviewer comment.
reviewer:
Comments 5: Authors must check the journal format.
- Comments 6: Authors must concentrate the clarity of the figures.
- Comments 7: Authors add five keywords in the manuscript.
- Comments 9: Authors add the line numbers and page numbers.
- Comments 10: Authors must add some topics in the section 2.
- Comments 11: Authors must check the grammar throughout the manuscript.
authors reply:
all the points have been addressed in the manuscript.
Round 2
Reviewer 1 Report
I would like to note significant changes in the text and content of the article. Changes in the text of the article led to its qualitative improvement. Definitely, the text of the article has become much better and clearer to readers. The article may be recommended for publication in the journal.
Author Response
thank you very much
Reviewer 2 Report
1. The authors mixed the numerical simulation within the Results and Discussion: e.g., the text on page 7 and page 10, where the authors described the hybrid model and the test cycle. This text is not part of the results or discussion and should belong to the methods. Please also check the other paragraphs.
2. Still, I think that you should explain your results rather than just illustrate them because you are working on a scientific paper rather than just a report.
3. Did any other studies conduct similar comparisons of what you have done? If so please compare your results with the others in detail. And I hope this comparison could address the Originality / Novelty of your study because now the Originality / Novelty needs improvements.
4. I doubt whether it’s appropriate to calibrate the hydrogen model with the data from other fuels.
Author Response
reviewer:
- The authors mixed the numerical simulation within the Results and Discussion: e.g., the text on page 7 and page 10, where the authors described the hybrid model and the test cycle. This text is not part of the results or discussion and should belong to the methods. Please also check the other paragraphs.
authors reply:
the text referring to the hybrid model and test cycle has been put in the "numerical simulations" section, all paragraphs have been checked
reviewer:
- Still, I think that you should explain your results rather than just illustrate them because you are working on a scientific paper rather than just a report.
authors reply:
the results have been better explained in the conclusion section, the reasons behind the increment of engine efficiency have been better highlighted
reviewer:
3. Did any other studies conduct similar comparisons of what you have done? If so please compare your results with the others in detail. And I hope this comparison could address the Originality / Novelty of your study because now the Originality / Novelty needs improvements.
authors reply:
now the results of this study have been compared with literature in the "conclusions" section and the Originality / Novelty of this stydy has been highlighted in "introduction"
reviewer:
4. I doubt whether it’s appropriate to calibrate the hydrogen model with the data from other fuels.
authors reply:
a lot of references reported in the manuscript refers to experimental tests, performed by the same author on the real engine object of the present study, when fueled with gasoline, natural gas and propane both in the naturally aspirated and in the supercharged configuration with volumetric compressor; this huge amount of experimental data allowed to accurately calibrate the 0D numerical model used in this work in many different operating conditions and to predict the gasoline engine BTE map. The hydrogen fueled engine maps have been obtained from the calibrated model making some hypothesis (the shorter combustion duration for example) that must be indeed experimentally validated but has been confirmed by some preliminary experimental tests, not yet published, by the same author as reported in the text (it is the future development of this work); the author believes that the results of the comparison between gasoline and hydrogen hybrid engines will not be greatly affected and remain valid. Other than this the ability of hydrogen to run on engines with excess air ratio of 2 and supercharging pressure around 2 bar without knocking or producing NOx is widely reported in the literature that is referenced in this work so the hypothesis made in the present work are widely acceptable.
Reviewer 3 Report
The author performed an acceptable revision.
Author Response
thank you very much
Reviewer 4 Report
I have checked the response to reviewer comments. I came to know that authors just randomly mention that "all the points have been addressed in the manuscript". There is no proper justification in the response file. So I don't want to give the positive feedback about the article.
Author Response
I am sorry but I think I have addressed all the reviewer comments except for the request of inserting the reference to some works that were judged inconsistent with the topics of the present work.
Round 3
Reviewer 2 Report
Please avoid References in Section Conclusion. You could move this to Section Results and Discussion.
Author Response
reviewer comment:
Please avoid References in Section Conclusion. You could move this to Section Results and Discussion.
authors reply:
the references and relative comments have been moved to the Results and Discussion section.
Reviewer 4 Report
Authors answer the reviewer comments - Reviewer comments has been addressed in a single line. And also they are not giving proper answer about comment 8.
Author Response
reviewer comment:
Comments 5: Authors must check the journal format.
author reply:
the paper has been formatted according to the journal format
reviewer comment:
- Comments 6: Authors must concentrate the clarity of the figures.
authors reply:
the figures and their captions have been re-arranged in order to be more clear
reviewer comment:
- Comments 7: Authors add five keywords in the manuscript.
authors reply:
in the authors opinion the current number of keyword is enough to highlight the main topics of the paper and it is not necessary to add more keywords, maybe the reviewer has some specific suggestions?
reviewer comment:
- Comments 8: Authors must add the following references
o https://doi.org/10.1016/j.ijhydene.2019.03.131
o https://doi.org/10.1016/j.fuel.2020.118120
o https://doi.org/10.1016/j.fuel.2020.117645
o https://doi.org/10.1007/s13204-021-01781-z
o https://doi.org/10.1016/j.scitotenv.2020.142389
authors reply:
the mentioned references are off topic with respect to the present paper which is focused on hydrogen fueled hybrid vehicles and not on hydrogen production tecniques.
reviewer comment:
- Comments 9: Authors add the line numbers and page numbers.
authors reply:
the page numbers are already present in the paper while the line numbers are not a part of the format requirements and this is the reason why the author did not add them.
reviewer comment:
- Comments 10: Authors must add some topics in the section 2.
authors reply:
section 2 has been widely re-arranged and new topics have been added: the description of the hybrid vehicle configuration and the description of the driving cycle implementation for both the hybrid and the conventional vehicle.
reviewer comment:
- Comments 11: Authors must check the grammar throughout the manuscript.
authors reply:
the grammar has been checked throughout the manuscript.